# ARIADNE: ADVANCING REAL-WORLD PATH-FINDING CAPABILITIES OF VLMS VIA DIFFICULTY-AWARE REINFORCEMENT LEARNING

## ABSTRACT

Recent advances have seen Vision-Language Models (VLMs) achieve impressive reasoning capabilities, largely demonstrated on tasks like mathematical problem solving via reinforcement learning. However, whether such methods can extend the fundamental reasoning bounds of VLMs to out of distribution complexities remains an underexplored question, as the cumulative and interconnected nature of knowledge in domains like mathematics makes it difficult to create truly isolated training and testing splits. To address this, we investigate multistep spatial reasoning, a domain where task difficulty can be systematically controlled. We introduce **Ariadne**, a training and evaluation framework centered on pathfinding puzzles where complexity is precisely defined by path length and turn count. This allows us to train on a curriculum of simpler puzzles and evaluate generalization on quantifiably harder, unseen tasks (e.g., training on paths with ≤3 steps and testing on paths with ≥5 steps). Our experiments reveal that while a strong base model like Qwen-VL-7B-Instruct fails on paths longer than two steps, our model, trained with RLVR, successfully generalizes to solving five step puzzles unseen during training. This result demonstrates that reinforcement learning can genuinely extend the intrinsic reasoning capabilities of VLMs. Surprisingly, although trained exclusively on synthetic mazes, Ariadne demonstrates performance gains on real world benchmarks like MapBench and ReasonMap, showcasing that core spatial reasoning skills transfer effectively even when the visual inputs, from simple mazes to complex real world maps, are entirely distinct.

## 1 INTRODUCTION

Reinforcement learning (RL) has played a pivotal role in boosting the reasoning capabilities of large language models (LLMs) Ouyang et al. (2022); Schulman et al. (2017). DeepSeek-R1 recently advanced this progress by showing that group relative policy optimization (GRPO), even when combined with simple rule-based rewards and no separately trained reward model, can effectively enhance LLMs with complex reasoning skills DeepSeek-AI et al. (2025); Shao et al. (2024). By leveraging tasks with deterministic ground-truth answers, this R1-like paradigm delivers precise and stable rewards DeepSeek-AI et al. (2025). In the vision-language domain, path-finding maze puzzles are particularly well-suited for such strategies due to their deterministic ground-truth annotations, making them ideal for rule-based reinforcement Dao & Vu (2025); Mirowski et al. (2017). Motivated by these insights, it is natural to investigate whether a similar GRPO-based framework can enhance the real-world path reasoning performance of vision-language models (VLMs) Jiaqi et al..

To further investigate the path reasoning capabilities of the GRPO-fine-tuned model under varying levels of problem complexity, we systematically controlled the difficulty of path-finding puzzles. We introduce **Ariadne**, a controllable GRPO framework that allows fine-grained manipulation of path-finding complexity while maintaining coherent reasoning structures and verifiable rewards, thereby enabling a more robust and effective reinforcement learning pipeline. To further understand the reasoning behavior of the model fine-tuned with GRPO, we also performed controlled manipulation of problem complexity on the in-domain puzzle test set. Specifically, we probed the model's reasoning mechanisms by varying the length of the ground-truth path and the number of turns it contains.

Our empirical investigation yields several key findings about puzzle-based GRPO: 1) Although the model is fine-tuned using trajectories with limited steps and turns, its reasoning capabilities generalize well to more complex scenarios requiring significantly longer paths or more frequent turns. 2) However, beyond a certain complexity threshold, the model completely fails to identify correct solutions, even though these problems remain well within the token length limits. Subsequently, we evaluate the model's reasoning improvement on out-of-domain data: real-world map navigation test sets. The results further confirm the effectiveness of GRPO on puzzles in enhancing generalizable reasoning performance.

Our findings reveal both the capabilities and constraints of current VLMs, prompting further examination of the underlying nature of spatial reasoning in these models. The main contributions of this work are:

- We propose Ariadne, a controllable GRPO training framework, curate a complexity-graded path-finding dataset, and systematically explore the generalization ability of VLMs in spatial reasoning, demonstrating their capacity to transfer from simple to increasingly complex puzzles.

- We evaluate Ariadne on out-of-distribution tasks with novel visual inputs and real-world map navigation settings, showing that GRPO-enhanced reasoning not only generalizes across domains but also exposes critical failure modes beyond a threshold of problem complexity.

## 2 RELATED WORK

### 2.1 REASONING IN VISION–LANGUAGE MODELS

VLMs have achieved substantial progress on tasks such as visual question answering (VQA) and image captioning Alayrac et al. (2022); Liu et al. (2023); Zhu et al. (2025). To strengthen reasoning capabilities, prior work has explored chain-of-thought (CoT) prompting Wei et al. (2022) and the construction of supervised fine-tuning (SFT) datasets enriched with step-level rationales Zhu et al. (2025). While these strategies enhance reasoning to some extent, they fall short of capturing human-like cognitive processes such as questioning and self-verification. Consequently, their effectiveness diminishes on complex reasoning tasks. Recent models such as DeepSeek-R1 Shao et al. (2024) attempt to address this gap by combining cold-start initialization with RL, thereby acquiring higher-quality multimodal CoT reasoning and achieving state-of-the-art results on challenging visual reasoning benchmarks.

### 2.2 REASONING WITH REINFORCEMENT LEARNING

Despite the advances in VLMs, spatial and multi-step reasoning, particularly in navigation and spatial understanding remains a key challenge for VLMs. To overcome this limitation, researchers have increasingly turned to RL-based approaches. Ji et al. applied GRPO with structured CoT supervision to spatial VQA and navigation tasks, showing that verifiable, rule-based rewards can significantly improve reasoning robustness Ji et al. (2025). Similarly, CoT-VLA extended CoT reasoning to vision–language–action models, where explicit intermediate reasoning steps yielded strong gains in robotic navigation Zhao et al. (2025). In synthetic navigation domains, Mirowski et al. enhanced RL training with auxiliary tasks such as depth prediction and loop-closure detection, enabling efficient traversal of complex 3D mazes Mirowski et al. (2017).

While these approaches have demonstrated effectiveness, most rely on large-scale synthetic data generation or are narrowly tailored to specific architectures. Consequently, open questions remain about how VLMs adapt their reasoning strategies when systematically exposed to increasing task complexity, and to what extent such improvements generalize beyond synthetic environments Feng et al. (2025); Xing et al. (2025).

In this work, we introduce a controllable GRPO training framework, Ariadne, for spatial path reasoning. Ariadne enables precise manipulation of problem complexity while preserving consistent logical structures and verifiable rewards, allowing for both targeted capability enhancement and

analysis of generalization and failure modes. Finally, through evaluation on real-world map navigation datasets, we demonstrate that reasoning gains achieved in synthetic puzzle environments can effectively transfer to practical, out-of-domain scenarios.

# 3 METHODOLOGY

## 3.1 PRELIMINARIES OF GRPO

Recent advancements in enhancing the reasoning capabilities of vision-language models (VLMs) have demonstrated that RL is a powerful training strategy Shao et al. (2024). In this study, we adopt GRPO as our learning framework. GRPO operates by directly comparing groups of candidate responses, thereby eliminating the need for a separate critic model.

During training, GRPO generates a set of candidate outputs $o_1, o_2, \ldots, o_G$ for a given question $q$ drawn from the dataset $\mathcal{D}$ using the old policy $\pi_{\theta_{\text{old}}}$. Each candidate response $o_i$ is then evaluated using a reward function $R(o_i, q)$, yielding a reward $r_i$. To assess the relative quality of the responses within the group, GRPO normalizes the rewards by computing their mean and standard deviation. The advantage $A_i$ for each response is then calculated as:

$$A_i = \frac{r_i - \text{mean}\{r_1, r_2, \ldots, r_G\}}{\text{std}\{r_1, r_2, \ldots, r_G\}} \tag{1}$$

GRPO optimizes the policy model $\pi_\theta$ by maximizing the following objective:

$$\mathcal{J}_{\text{GRPO}}(\theta) = \mathbb{E}_{q \sim \mathcal{D}, \{o_i\}_{i=1}^G \sim \pi_{\theta_{\text{old}}}(O|q)}$$

$$\left[ \frac{1}{G} \sum_{i=1}^G \min \left( \frac{\pi_\theta(o_i|q)}{\pi_{\theta_{\text{old}}}(o_i|q)}, \text{clip}\left( \frac{\pi_\theta(o_i|q)}{\pi_{\theta_{\text{old}}}(o_i|q)}, 1-\epsilon, 1+\epsilon \right) \right) \right] \tag{2}$$

where $\epsilon$ is the clipping hyper-parameter.

## 3.2 REWARD FUNCTION

We design a reward function to measure the stepwise correctness of model-generated answers relative to a ground truth reference. The function (Algorithm 1) assigns proportional rewards for both fully correct and partially correct answers, using the number of reasoning *turns* as a scaling factor.

**Step Extraction.** Each model completion is first standardized via a format extraction function to ensure consistent formatting. Reasoning steps are represented as a sequence of *moves* embedded in the format:

$$<|\text{step content}|>.$$

A count turns function extracts these moves using a regular expression, returning both the ordered list of moves and the number of *turns*, defined as the count of transitions between consecutive distinct moves.

**Reward Calculation.** Let $R = [r_1, r_2, \ldots, r_m]$ be the predicted move sequence and $A = [a_1, a_2, \ldots, a_n]$ be the ground truth. The reward is computed as:

$$\text{reward} = \begin{cases} 0.2 \times m \times \text{turns}(A), & \text{if } R = A, \\ 0.1 \times k \times \text{turns}(A_{1:k}), & \text{if only first } k \text{ moves match.} \end{cases}$$

This ensures:

- Full matches receive maximum proportional reward.
- Partial matches are rewarded according to the matching prefix length $k$.
- The conversational or reasoning complexity, measured by *turns*, scales the reward.

---

**Algorithm 1** Correctness Reward Function

---

1: **function** CORRECTNESS REWARD(completions, answers)
2:     rewards ← []
3:     **for all** $r, a$ in (completions, answers) **do**
4:         $r\_moves, r\_turns$ ← count_turns($r$)
5:         $a\_moves, a\_turns$ ← count_turns($a$)
6:         **if** $r\_moves = a\_moves$ **then**
7:             reward ← len($a\_moves$) $\times 0.2 \times a\_turns$
8:         **else**
9:             $k$ ← length of matching prefix($r\_moves, a\_moves$)
10:        $k\_turns$ ← count_turns(first $k$ moves of $a$)
11:        reward ← $k \times 0.1 \times k\_turns$
12:        **end if**
13:        rewards.append(reward)
14:     **end for**
15:     **return** rewards
16: **end function**

---

## 4 EXPERIMENTS

### 4.1 EXPERIMENTAL SETTINGS

#### 4.1.1 BASE MODEL AND IMPLEMENTATION

We adopt Qwen2.5-VL-7B-Instruct Bai et al. (2025) as our policy model due to its strong capabilities in vision-language understanding, which we aim to further improve via reinforcement learning. In our Ariadne framework, Qwen2.5-VL-7B-Instruct is fine-tuned using approximately 2,000 samples from the AlphaMaze dataset with the GRPO algorithm, targeting enhanced reasoning performance in maze navigation tasks. Our reward function combines three factors: answer accuracy, answer format, and reasoning format, emphasizing correctness while maintaining response clarity and structure. Training is conducted on 8 NVIDIA A100 40GB GPUs with a learning rate of 1e-6, 1 iteration, batch size of 1 per device, and 16 gradient accumulation steps. We apply a warmup ratio of 0.05, and for each prompt, 8 candidate responses are sampled using a temperature of 1.0.

#### 4.1.2 DIFFICULTY CONTROL

To control the difficulty of training samples, we design the distribution of maze step counts $s \in \{1, 2, 3, 4, 5\}$ based on an inverted Gaussian-like distribution centered at step count 3. We posit that frequent exposure to simple cases, where each step involves selecting from up to four possible directions but is often constrained by walls blocking some paths, helps the model acquire stable and generalizable low-level navigation patterns by learning to effectively choose feasible moves in a limited and structured action space. At the same time, challenging cases (with longer trajectories) drive the model to learn global spatial reasoning and coherent path planning over multiple steps. Specifically, the sampling probability for each step count is defined as:

$$P(s) \propto 1 - \exp\left(-\frac{(s-\mu)^2}{2\sigma^2}\right), \tag{3}$$

where $\mu = 3$ is set as the midpoint of the step range and $\sigma = 2$ is manually chosen to control the spread. This design ensures that both simpler (1–2 steps) and more complex trajectories (4–5 steps) are sampled more frequently. The resulting empirical distribution approximates $\{21\%, 18\%, 16\%, 18\%, 21\%\}$ respectively. Additionally, a test set is constructed from AlphaMaze by evenly sampling across the number of moves.

Figure 1 illustrates the statistical properties of navigation trajectories in both the training and testing sets. Panels (A) and (C) depict the distributions of step lengths, reflecting the frequency of various movement distances during training and testing episodes, respectively. Panels (B) and (D) show

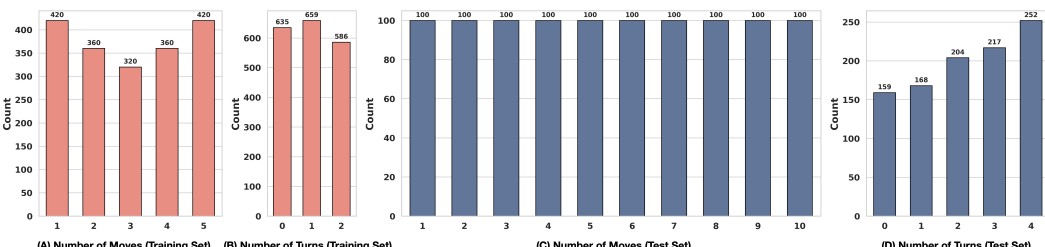

Figure 1: (A) Step length distribution in the AlphaMaze training set, where the number of moves $s \in \{1, 2, 3, 4, 5\}$ is sampled according to an inverted Gaussian-like distribution centered at $s = 3$, ensuring higher frequencies for both simple and complex cases. (B) Distribution of directional turns in the training set under the same controlled sampling scheme. (C) Step length distribution in the AlphaMaze testing set, constructed via uniform sampling. (D) Distribution of directional turns in the testing set under the same controlled sampling scheme.

the distributions of directional turns, emphasizing the complexity of orientation changes involved in path-finding across the two sets.

During training, the number of moves is limited to 1–5 and the number of turns to 0–2. In the testing phase, these constraints are relaxed to 1–10 moves and 0–4 turns, allowing for a more comprehensive evaluation of the model's path-finding capabilities, acquired through GRPO on mazes, and its ability to generalize navigation behavior to more complex yet still in-domain scenarios.

### 4.1.3 PROMPT TEMPLATE

---

**System Prompt**

You are a navigation assistant to solve visual path-finding tasks.
Your goal is to infer a valid path from a visually marked starting point (green cell labeled 'O') to a visually marked target (red cell labeled 'T') by analyzing the maze image.
Rules:
- The maze is composed of open paths and impassable black walls.
- Movement is only allowed through open paths, not through walls.
- You can move one step at a time in the four cardinal directions: `<|up|>`, `<|down|>`, `<|left|>`, `<|right|>`.
Output Format:
Think through each step inside `<think>` and `</think>` tags.
At each step:
1. Describe your current position based on visual layout and structure (e.g.,"in a corridor", "facing a wall", "at a crossroad", "turning a corner").
2. Decide the next move, and explain your reasoning.
3. Move and continue the path.
After your full reasoning, output only the final movement sequence using the allowed tokens:
`<|up|><|down|><|left|><|right|>`

---

### 4.2 BENCHMARKS AND METRICS

As illustrated in Figure 2, we utilize MapBench and ReasonMap to evaluate the path-finding capabilities of open-source MLLMs. MapBench consists of human-readable, outdoor navigation tasks curated from challenging real-world scenarios Feng et al. (2025); Xing et al. (2025). ReasonMap assesses fine-grained visual understanding through high-resolution transit maps from global cities such as Los Angeles, Toronto, and Beijing. It adopts a two-tier evaluation approach, comprising short and long questions, to measure both answer correctness (via accuracy) and answer quality (via a proposed map score that reflects route feasibility and efficiency).

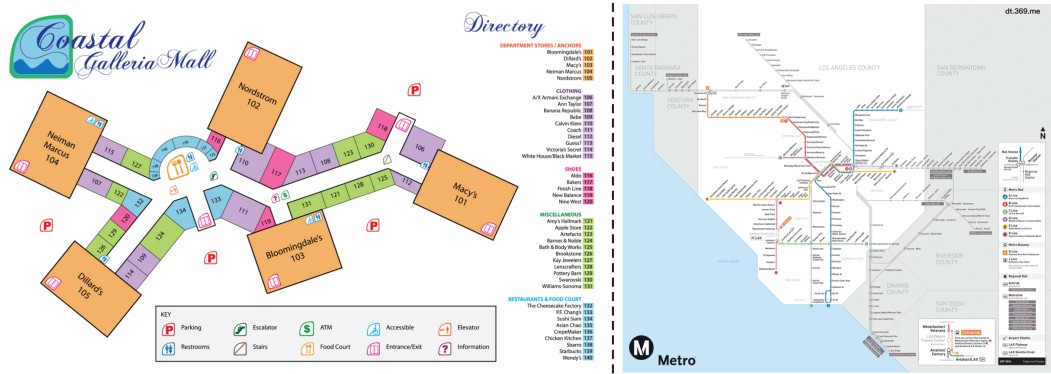

Figure 2: Illustrative examples from our two controlled benchmarks for path-finding evaluation. **MapBench** (left) features human-readable, outdoor navigation tasks derived from challenging real-world scenarios (e.g., mall navigation), designed to assess naturalistic instruction-following and local decision-making. **ReasonMap** (right) uses high-resolution transit maps from global metropolitan systems (e.g., Los Angeles, Toronto, Beijing), with a two-tier evaluation (short vs. long questions) to probe both fine-grained visual comprehension and global route planning.

In MapBench, models are tasked with generating language-based navigation instructions given a map image and a query specifying the starting and ending landmarks. The path quality score is defined as the ratio between the length of the model-generated path and the length of the ground-truth shortest path, serving as a metric for route efficiency.

For ReasonMap, accuracy is assessed through a comprehensive validation process that includes verifying the departure and arrival stops, matching each segment's route name with the map metadata, confirming the validity of intermediate stops, and ensuring transfer consistency. An answer is considered correct only if all criteria are met. For short questions, the map score emphasizes consistency in route and endpoints. For long questions, it additionally accounts for the number of stops and the correctness of specific via stops. Both accuracy and map score are weighted based on question and map difficulty, enabling difficulty-aware evaluation.

### 4.3 MAIN RESULTS

#### 4.3.1 MODEL GENERALIZATION ON COMPLEX IN-DOMAIN TASKS FOLLOWING GRPO TRAINING

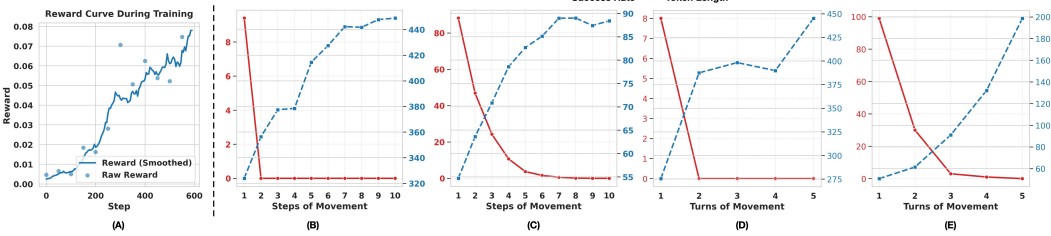

Figure 3: Training reward dynamics and evaluation of path-following ability. (A) Reward curve during GRPO training, showing steady improvement of rewards and stable learning progress. (B, D) For Qwen2.5-VL-7B-Instruct, the success rate collapses rapidly to near zero as movement steps or directional turns increase, while token length steadily grows, suggesting the model generates longer but largely unsuccessful trajectories. (C, E) In contrast, our Ariadne maintains substantially higher success rates, with performance declining more gradually as path complexity increases. Token length grows moderately, reflecting stronger robustness and better generalization to longer and more complex navigation tasks compared with Instruct.

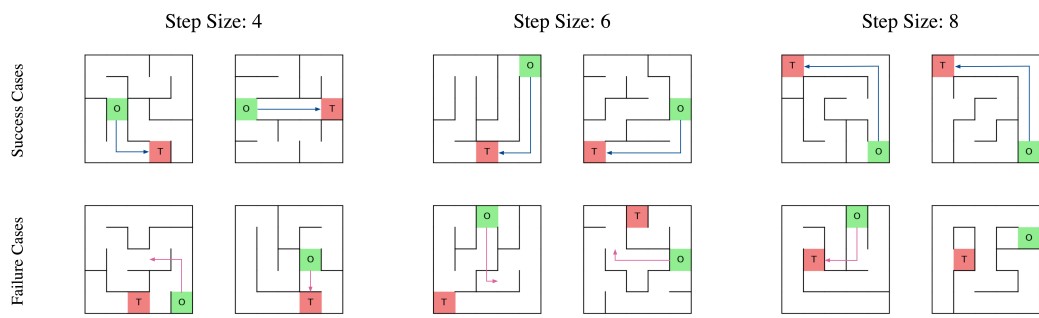

Figure 4: Representative success (top row) and failure (bottom row) cases from the AlphaMaze test set under controlled step sizes (4, 6, 8). Success cases generally correspond to smoother layouts with limited detour requirements, enabling coherent long-range navigation. Failure cases, by contrast, arise in locally complex structures characterized by dense turns, narrow passages, and elongated detours, which challenge the model's ability to maintain global path consistency.

Figure 3 presents the reward trajectory during GRPO training on AlphaMaze, together with quantitative evaluations on the test set. Across all evaluated movement-step ranges, Ariadne consistently outperforms its base policy model, Qwen2.5-VL-7B-Instruct. Notably, the performance margin between the two models becomes more pronounced as task complexity increases, particularly in cases requiring substantially longer solution paths. In low-turn conditions (0 or 1 turn), our Ariadne also maintains a clear advantage, achieving higher success rates across the board. Nevertheless, as the number of directional turns grows, both models experience a sharp accuracy decline, suggesting that high-turn mazes constitute a shared failure mode. This pattern indicates that while GRPO significantly enhances generalization to longer trajectories, it only partially alleviates the difficulties posed by intricate path geometries.

In addition, the qualitative cases in Figure 4 provide additional insight into the sources of failure. Successful trajectories typically occur in mazes with relatively smooth layouts and moderate detour requirements, where Ariadne enables coherent navigation over extended distances. In contrast, failed trajectories are strongly associated with local structural complexity: dense geometric patterns with frequent directional shifts, narrow passages that constrain feasible moves, and elongated detours that require maintaining global path consistency. Such observations underscore that the remaining challenges lie not in token capacity but in handling the combinatorial explosion introduced by sharp turns and tightly constrained maze topologies.

### 4.3.2 PERFORMANCE EVALUATION ON OUT-OF-DOMAIN REAL-WORLD NAVIGATION TASKS

As shown in Figure 5, the two models exhibit a clear divergence in real-world map-based navigation across diverse task settings. The baseline Qwen2.5-VL-7B-Instruct frequently suffers from systematic localization and planning errors, including misidentifying target positions, selecting inefficient detours, and ignoring environmental constraints such as rivers, enclosed building walls, or other impassable barriers. These shortcomings often result in trajectories that are either incomplete or infeasible. In contrast, Ariadne demonstrates a markedly more robust navigation strategy. By adopting a fine-grained, node-by-node planning mechanism, it maintains consistent goal-directed progress while respecting structural boundaries. Crucially, this approach generalizes across both unstructured, curvilinear outdoor networks (e.g., trails) and structured, grid-like indoor layouts (e.g., museums), highlighting the effectiveness of GRPO-induced adaptations beyond synthetic maze environments.

To further assess the transferability of these navigation gains to broader reasoning abilities, we evaluate both models on ReasonMap, a benchmark probing stepwise reasoning under varying difficulty levels (Figure 6). The results show that Ariadne delivers consistent gains in long-reasoning regimes, particularly in higher-complexity categories such as easy–middle and easy–hard, which correspond to question complexity and map complexity, respectively. For instance, in the easy–hard category, it achieves both a higher average score (5.17 vs. 4.65) and accuracy (6.67% vs. 5%) compared to

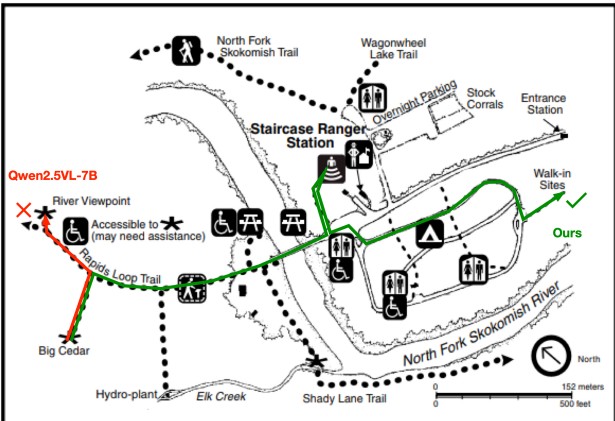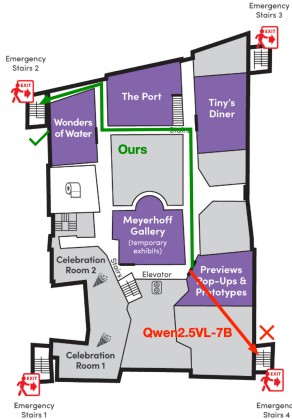

Figure 5: Representative path-finding results on MapBench for Trail (left, unstructured outdoor layout) and Museum (right, structured indoor layout) tasks. The baseline Qwen2.5-VL-7B-Instruct (red) often produces incomplete or infeasible trajectories due to systematic localization and planning errors, such as misidentifying targets, inefficient detours, and violations of environmental constraints. In contrast, Ariadne (green) leverages fine-grained node-by-node planning to sustain coherent, goal-directed progress while respecting map boundaries.

the baseline. These gains are not observed in short-reasoning settings, where performance between the two models remains comparable. Taken together, these findings suggest that the Maze-specific GRPO training paradigm not only enhances real-world navigation robustness but also strengthens the model's capacity for multi-step, long-horizon reasoning, an ability directly relevant for generalizing beyond controlled synthetic tasks.

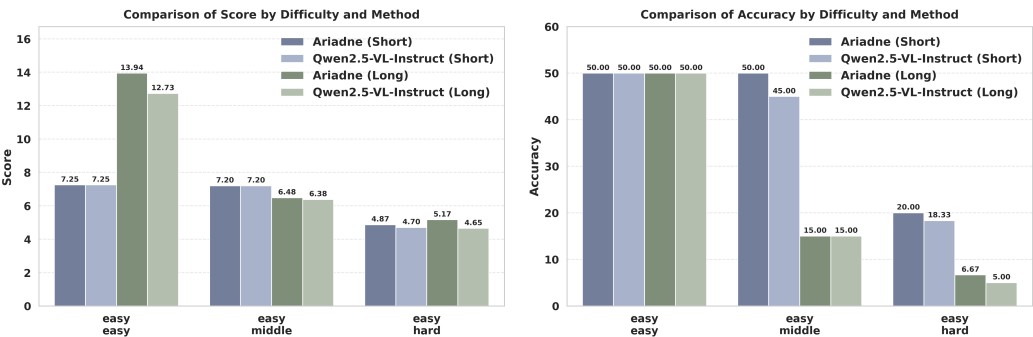

Figure 6: Performance comparison on ReasonMap across increasing question difficulty levels. (Left) Accuracy results indicate that Ariadne achieves consistent gains in long-reasoning categories (e.g., easy–middle, easy–hard), where the baseline Qwen2.5-VL-7B-Instruct struggles. (Right) Average map scores show a similar trend, with Maze-specific GRPO training improving solution feasibility and efficiency.

Table 1: Performance comparison on MapBench. **Bold** indicates the best performance; underline denotes the second best.

| Model | Google Map ↓ | Mall ↓ | Museum ↓ | National Park ↓ | Theme Park ↓ | Trail ↓ | Campus ↓ | Urban ↓ | Zoo ↓ |
|---|---|---|---|---|---|---|---|---|---|
| Qwen2-VL-7B-Instruct | 2.16 | 2.02 | 1.92 | 2.07 | 2.31 | 2.41 | 2.56 | 2.60 | 2.34 |
| LLaMA-3.2-11B-Vision-Instruct | 2.68 | 3.58 | 3.18 | 2.50 | 3.05 | 2.91 | 6.43 | 2.99 | 3.13 |
| InternVL3-VL-8B-Instruct | 2.82 | 3.97 | 2.37 | 3.12 | 2.63 | 2.97 | 2.23 | 2.70 | 2.31 |
| Qwen2.5-VL-7B-Instruct | 2.15 | 2.02 | 1.78 | 2.17 | 2.05 | 2.32 | **2.02** | 2.61 | **2.22** |
| Ariadne | **1.99** | **1.90** | **1.59** | **1.90** | **1.92** | **2.03** | 2.03 | **1.89** | 2.25 |

Table 2: Performance comparison on ReasonMap. $S$ = short questions, $L$ = long questions. **Bold** indicates the best performance; underline denotes the second best.

| Model | Weighted Acc. $(S)\uparrow$ | #Tokens $(S)$ | Weighted Acc. $(L)\uparrow$ | #Tokens $(L)$ | Weighted Map Score $(S / L)\uparrow$ |
|---|---|---|---|---|---|
| Qwen2-VL-7B-Instruct | 11.13% | 34 | 7.91% | 274 | 3.47 / 4.32 |
| LLaMA-3.2-11B-Vision-Instruct | 1.02% | 94 | 0.29% | 103 | 0.59 / 0.52 |
| InternVL3-VL-8B-Instruct | 6.30% | 36 | 4.54% | 67 | 2.45 / 3.44 |
| Qwen2.5-VL-7B-Instruct | **13.32**% | 26 | 6.00% | 61 | 3.73 / 4.51 |
| Ariadne | 13.03% | 27 | **6.59**% | 60 | **3.74 / 4.71** |

Overall, despite the intrinsic challenges posed by fine-grained visual perception and spatial reasoning, Ariadne achieves state-of-the-art performance on both MapBench and ReasonMap (Tables 1 and 2). The model demonstrates robust generalization across heterogeneous spatial layouts, encompassing both unstructured outdoor networks and structured indoor grids, while also exhibiting clear advantages in tasks requiring long-horizon, multi-turn reasoning. Such improvements are especially salient under conditions of elevated path complexity and extended reasoning chains, where conventional instruction-tuned baselines degrade sharply. These findings collectively confirm that GRPO training on the synthetic maze dataset substantially strengthens spatial-visual comprehension, yielding a model that stands at the forefront of its scale class for advanced spatial reasoning and real-world navigation tasks.

## 5 Discussion and Conclusion

In this work, we systematically investigated the spatial reasoning capabilities of GRPO-fine-tuned vision-language models within our controlled complexity framework, Ariadne. We found that models trained on short, simple trajectories generalize surprisingly well to much longer and more intricate paths. Real-world map navigation experiments further demonstrated that the benefits of GRPO extend beyond synthetic puzzles, underscoring the broader applicability of our approach. However, this capability declines sharply once a critical complexity threshold is crossed. Notably, these failures occur even when problems remain within the token budget, suggesting that the limitation arises from the structure of multimodal reasoning rather than memory constraints.

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
