# OpenReview forum: "Ariadne: Advancing Real-world Path-finding Capabilities of VLMs via Difficulty-aware Reinforcement Learning"
_ICLR.cc/2026/Conference — ICLR 2026 Conference Withdrawn Submission_

### Official Review · Reviewer_txtD · 2025-10-24

**Soundness:** 1
**Presentation:** 1
**Contribution:** 1
**Rating:** 2
**Confidence:** 5

**Summary:**

This paper presents Ariadne, a reinforcement learning framework for training and evaluating multimodal large language models (MLLMs) on pathfinding puzzles. Ariadne is deliberately designed for controllability, allowing precise examination of the dynamics of RL-based MLLM training. Experimental results show that while current state-of-the-art vision-language models struggle with even simple maze-solving tasks, reinforcement learning enables them to generalize effectively to unseen and more challenging environments. Moreover, Ariadne achieves notable improvements on real-world benchmarks such as MapBench and ReasonMap, despite being trained exclusively on synthetic maze data.

**Strengths:**

1. The problem statement is clear: investigating multi-step spatial reasoning provides an effective lens for assessing MLLMs’ visual reasoning abilities that extend beyond conventional mathematical reasoning.
2. The out-of-domain evaluation on real-world map reasoning serves as a useful exploratory experiment, illustrating that RLVR trained purely on synthetic data can potentially transfer its learned spatial reasoning skills to complex, real-world settings.

**Weaknesses:**

1. Contribution is limited. The paper adopts an existing training algorithm (GRPO) and dataset (Alphamaze) without any modification or new insight, which is already done in the original Alphamaze paper. The only novelties are the new training dataset difficulty distribution and a difficulty-aware reward function. However, no ablation study is performed to demonstrate the effectiveness of the dataset distribution and reward function design.
2. Lack of thorough description of the research content. Before section4, there is no introduction or description of the training dataset used in the main experiment, making the design of the dataset distribution and the reward function confusing in section3.
3. The claimed performance gains on real-world dataset is not convincing. In Table2, Ariadne underperforms raw Qwen2-VL and Qwen2.5VL in two metrics respectively, still lagging significantly behind the strong proprietary models like OpenAI-o3.

Overall, this paper fails to deliver novel insights or contributions to the research community. The proposed methods and design choices are inadequately motivated and lack rigorous justification. Furthermore, the experimental results do not convincingly support the claimed contributions. In its current form, the paper does not meet the standards of originality, methodological rigor, or empirical validity required for publication. I therefore recommend rejection.

**Questions:**

1. It is important to elaborate the difference of this work with the training in AlphaMaze
2. What is the motivation of the Gaussian-like distribution of the training dataset? Have the author try to compare it with normal distribution?
3. It is also important to include more details about how the trained models solve the out of domain map dataset, is there any behavior shift?

---

### Official Review · Reviewer_8H4T · 2025-10-29

**Soundness:** 2
**Presentation:** 1
**Contribution:** 2
**Rating:** 2
**Confidence:** 4

**Summary:**

This paper introduces Ariadne, a training and evaluation framework that uses reinforcement learning (specifically GRPO - Group Relative Policy Optimization) to enhance the spatial reasoning and path-finding capabilities of Vision-Language Models (VLMs). The key innovation is systematic difficulty control through path length and turn count, enabling rigorous evaluation of whether RL can genuinely extend VLM reasoning capabilities to out-of-distribution complexities.

**Strengths:**

This paper is a nice practice of applying GRPO on an existing dataset (AlphaMaze) and testing on existing benchmarks (MapBench and ReasonBench).

**Weaknesses:**

**1. Limited Technical Contribution**

i. The paper's core technical contribution is limited, primarily consisting of a reward function design (Algorithm 1) and a difficulty-controlled sampling strategy for an existing dataset. The training methodology itself applies standard GRPO without modifications, and the evaluation uses existing benchmarks (MapBench, ReasonMap).

ii. More critically, the paper lacks ablation studies to validate its design choices. Specifically: (1) no ablation on the reward function components (why 0.2 vs 0.1 scaling factors? why scale by turn count?), (2) no comparison with simpler reward designs (e.g., binary correct/incorrect rewards), (3) no analysis of whether the difficulty-controlled sampling distribution is optimal compared to alternatives (e.g., uniform sampling, different Gaussian parameters), and (4) no ablation on prompt template design. Without these ablations, it is unclear which components are essential versus incidental to the reported improvements.

**2. Limited Dataset Transparency and Methodology**

i. The paper suffers from insufficient citation and unclear attribution of the AlphaMaze[1] dataset, which forms the foundation of the entire training framework. While AlphaMaze is briefly mentioned once in line 043, this single reference is inadequate given the dataset's central role in the work. Throughout the rest of the manuscript, including the methodology (Section 4.1.2) and results sections (Section 4.3.1), the authors refer to "AlphaMaze training set" and "AlphaMaze test set" without proper citations, creating ambiguity about whether this is an existing benchmark or a novel contribution.

ii. The paper omits essential information about the original AlphaMaze dataset, including its total size, construction methodology, and statistical properties. Readers cannot determine: (1) whether the ~2,000 selected samples represent the full dataset adequately, (2) how the authors' controlled difficulty distribution (Figure 1) compares to AlphaMaze's original distribution, or (3) what sampling biases may exist. This lack of transparency undermines both proper attribution to prior work and the reproducibility of the experimental design.

**3. Limited Comparisons**

i. Unfair baseline setup. Baselines (Qwen2.5-VL-7B-Instruct, LLaMA-3.2-11B, InternVL3-8B) are evaluated zero-shot, while Ariadne is trained on 2,000 AlphaMaze samples. This makes the comparison invalid and does not show GRPO is necessary. The below baselines are simpler but important to verify the proposed methods and at least some of them should be evaluated:

- SFT on the same 2,000 samples
- Few-shot prompting
- Distill on closed-source models' trajectories


ii. No evaluation of closed-source models. The paper omits comparisons with state-of-the-art proprietary models (i.e. GPT-4o) on both AlphaMaze and real-world benchmarks. It can be at least reference signals for your results.


**4. Misaligned results**

- Performance of Qwen2-VL-7B-Instruct and LLaMA-3.2-11B-Vision-Instruct  does not align with MapBench's[2] original paper. Can the authors answer why?


**5. Wrong citation format**

The author should use \citep for most of the citations in the paper.

[1] Dao, Alan, and Dinh Bach Vu. "AlphaMaze: Enhancing Large Language Models' Spatial Intelligence via GRPO." arXiv preprint arXiv:2502.14669 (2025).

[2] Xing, Shuo, Zezhou Sun, Shuangyu Xie, Kaiyuan Chen, Yanjia Huang, Yuping Wang, Jiachen Li, Dezhen Song, and Zhengzhong Tu. "Can Large Vision Language Models Read Maps Like a Human?." arXiv preprint arXiv:2503.14607 (2025).

**Questions:**

See Weaknesses.

---

### Official Review · Reviewer_AW46 · 2025-10-31

**Soundness:** 1
**Presentation:** 1
**Contribution:** 2
**Rating:** 2
**Confidence:** 4

**Summary:**

Using Group Relative Policy Optimization (GRPO) ARIADNE is trained on synthetic maze-based path-finding puzzles and evaluated on real-world map-navigation benchmarks such as MapBench and ReasonMap. The authors claim that Ariadne enables generalization from simple synthetic mazes to complex, real-world navigation tasks, demonstrating improved out-of-distribution reasoning capabilities and substantially strengthened spatial-visual comprehension.

**Strengths:**

ARIADNE shows improved performance on real-world out-of-distribution data after training on significantly different synthetic training data.

**Weaknesses:**

The experimental framework is interesting and technically plausible, but key elements (e.g., the incorrect GRPO formula, lack of ablations, and absence of multiple runs or confidence intervals) weaken the empirical support for the central claims. The current validation does not convincingly demonstrate that the observed gains arise from deeper reasoning improvements rather than pipeline or prompt robustness.

# Presentation
The GRPO formula (2), which is the main formula, is wrong (A_i missing). This pattern of lack of attention to detail and confusing presentation remains throughout the paper.

The readability of the figures is poor and they are hard to read in both formatting and content-wise. Examples include:

Figure 3:
- text size too small
- caption is hard to understand; difference between B & D is not directly apparent, same for C & E
- the range of the y-axis of B & D is comparable and it would be way easier to compare the two if the same limit was used, same for C & E
- maybe consider merging the plots so that the approaches can actually be compared and at the same time, this allows for bigger text size

Figure 6:
- “Accuracy (Left)” is wrong, the accuracy plot is on the right.



Minor:
- 352: “In addition, the qualitative cases in Figure 4 provide additional insight …” text fluency could be improved

# Method
Seperate runs, confidence intervals and an ablation study would strengthen the claims. As of now, the validation study of the generalization to real maps does not provide sufficient support for the claims made.

Figure 6 (right) has 5 entries at 50%, while this is possible, it is curious and should be addressed. It seems like it was evaluated on little data and with a single run. Furthermore on the left side the biggest difference is seen in the score for the easy-easy tasks, but this is not reflected in the accuracy for the easy-easy tasks on the right. Similar discrepancies are seen for the easy-hard tasks where the accuracy for the long versions goes down significantly. A more thorough and detailed validation analysis is required. The data presented in Figure 6 also raises questions about the effectiveness of the method, which needs to be addressed more strongly. “The results show that Ariadne delivers consistent gains in long-reasoning regimes, particularly in higher-complexity categories such as easy–middle and easy–hard, which correspond to question complexity and map complexity, respectively.” This is a strong statement considering that the scores do not reflect this. The scores that indicate the opposite should be more clearly addressed.

I remain sceptical on the origin of the gains in path planning skills. How much is purely due to improved robustness of the pipeline and understanding of the task+prompt vs. the claimed improvement in fundamental reasoning capabilities and substantial strengthening of spatial-visual comprehension: “This result demonstrates that reinforcement learning can genuinely extend the intrinsic reasoning capabilities of VLMs.” and “These findings collectively confirm that GRPO training on the synthetic maze dataset substantially strengthens spatial-visual comprehension, yielding a model that stands at the forefront of its scale class for advanced spatial reasoning and real-world navigation tasks.” While the authors provide a list of failure reasons for the baseline: “The baseline Qwen2.5-VL-7B-Instruct frequently suffers from systematic localization and planning errors, including misidentifying target positions, selecting inefficient detours, and ignoring environmental constraints such as rivers, enclosed building walls, or other impassable barriers.” There is no evidence supporting this or further explanations. There is a lack of information and validation to allow for a systematic attribution of capability improvements. Even then, the improvement on real-world benchmarks is interesting but narrow (map-based reasoning). It’s unclear how transferable these benefits are to broader VQA or embodied reasoning tasks.
Contribution
The contribution is overstated as mentioned above as well: “In this work, we systematically investigated the spatial reasoning capabilities of GRPO-fine-tuned vision-language models within our controlled complexity framework, Ariadne.”

The paper extends GRPO-based reasoning work to a path-finding domain. While the experimental setup is appealing and the results on real-world map data are promising, the conceptual novelty is limited. The contribution would be stronger if framed as a domain-specific study of GRPO generalization rather than as evidence of fundamentally new reasoning capabilities.

**Questions:**

Addressing the weaknesses and a more thorough and careful analysis of the generalization to the real maps would make this a valuable contribution.

---

### Note · Authors · 2025-11-12

I have read and agree with the venue's withdrawal policy on behalf of myself and my co-authors.